# Multi-modal Grouping Network for Weakly-Supervised Audio-Visual Video Parsing

**Shentong Mo**
Carnegie Mellon University

**Yapeng Tian**[*]
University of Texas at Dallas

## Abstract

The audio-visual video parsing task aims to parse a video into modality- and category-aware temporal segments. Previous work mainly focuses on weakly-supervised approaches, which learn from video-level event labels. During training, they do not know which modality perceives and meanwhile which temporal segment contains the video event. Since there is no explicit grouping in the existing frameworks, the modality and temporal uncertainties make these methods suffer from false predictions. For instance, segments in the same category could be predicted in different event classes. Learning compact and discriminative multi-modal subspaces is essential for mitigating the issue. To this end, in this paper, we propose a novel Multi-modal Grouping Network, namely MGN, for explicitly semantic-aware grouping. Specifically, MGN aggregates event-aware unimodal features through unimodal grouping in terms of learnable categorical embedding tokens. Furthermore, it leverages the cross-modal grouping for modality-aware prediction to match the video-level target. Our simple framework achieves improving results against previous baselines on weakly-supervised audio-visual video parsing. In addition, our MGN is much more lightweight, using only 47.2% of the parameters of baselines (17 MB vs. 36 MB). Code is available at https://github.com/stoneMo/MGN.

## 1 Introduction

Humans understand the surrounding environment by integrating signals from different senses. In our daily life, sound and sight are two of the most commonly used modalities, which have drawn much attention from researchers to explore computational audio-visual scene understanding.

Previous audio-visual work [1, 2] usually assumes audio and visual data are temporally aligned. However, the alignment does not always exist in real-world videos. For example, sounding objects in many videos are outside of the field-of-view (FoV). For these non-aligned cases, audio signals become more reliable in understanding the events of interest. In this work, we address the audio-visual video parsing (AVVP) task [3] that aims to parse a video into temporal event segments and predict the audible, visible, or audio-visible event categories.

Existing approaches [3, 4, 5] usually focus on learning to leverage the unimodal and cross-modal temporal contexts from weak supervisions. HAN [3] introduced a simple Multimodal Multiple Instance Learning framework with cross-modal and self-modal attention layers to utilize the video-level labels. Recent state-of-the-art methods usually use the HAN as the baseline and modify it to further improve parsing performance. Particularly, contrastive learning and label refinement are proposed in Wu and Yang [4], where they adopted a contrastive loss to enforce the temporal alignment between the audio and visual features at the same timestamp and augmented training data with modality-aware event labels generation. Furthermore, Lin *et al.* [5] proposed to leverage

---

[*]Corresponding author.

audio-visual class co-occurrence to jointly explore the relationship of different categories among all modality streams.

Our key motivation is to learn compact and discriminative audio and visual representations by explicit multi-modal grouping for mitigating the modality and temporal uncertainties in the weakly-supervised audio-visual video parsing problem. During training, segment-wise event labels are unavailable for audio and visual temporal segments in videos. Thus, the multi-modal temporal modeling in the above existing methods might aggregate and implicitly group irrelevant semantic information due to lacking of fine-grained supervisions, which causes false positives for predicting categories of events. That is, audio or visual temporal segments in the same event category might be far away from the class center in the embedding space since there are no segment-level and modality-wise constraints during training. In the meanwhile, there is no constraint for modality category prediction to match the video-level target at the end.

Different from past approaches, we propose a new Multi-modal Grouping Network, namely MGN, to explicitly group semantic-aware multi-modal contexts, which enables learning more compact and discriminative audio and visual representations. Specifically, we first extract event-aware unimodal features through unimodal grouping in terms of learnable categorical embedding tokens for each individual modality. Then, we introduce a cross-attention layer with a hard attention mechanism to aggregate cross-modal temporal contexts. Finally, we utilize a cross-modal grouping module to predict the modality category from updated class-aware unimodal embeddings.

Experimental results on the LLP [3] dataset validate that our new audio-visual video parsing framework achieves superior results over previous state-of-the-art methods [1, 2, 3, 4]. Empirical results also demonstrate the generalizability of our approach to contrastive learning and label refinement proposed in MA [4]. In addition, we substantially reduce the parameters of previous work by using only 47.2% parameters of baselines (17 MB vs. 36 MB).

Our main contributions can be summarized as follows:

- We propose a new audio-visual video parsing baseline: Multi-modal Grouping Network (MGN) that enables explicit grouping in a multi-modal network to learn compact and discriminative audio and visual embeddings.

- We introduce class-aware unimodal grouping and modality-aware cross-modal grouping modules to aggregate multi-modal temporal contexts.

- The experiments can demonstrate the superiority of our MGN over state-of-the-art AVVP approaches and its generalizability to contrastive learning and label refinement.

## 2    Related Work

**Audio-Visual Learning.** Audio-visual learning has addressed in many previous works [6, 7, 8, 9, 10, 11, 12, 13, 14, 15, 16] to learn the audio-visual association between the two distinct yet correlated modalities. Such audio-visual temporal association is crucial to several tasks, such as audio-visual spatialization [17, 18, 19, 14], speech/audio separation [13, 20, 21, 11, 12, 22, 23, 24], visual sound source localization [10, 25, 26, 27, 28, 29]. A comprehensive survey [30] covers recent advances in audio-visual learning. In this work, we mainly focus on audio-visual video parsing that aims to parse a video into temporal segments in terms of audio, visual, and audio-visual events with only video-level annotations for training. In addition, different from existing audio-visual architectures, we firstly introduce explicit multi-modal grouping into audio-visual learning.

**Audio-Visual Video Parsing.** Audio-visual video parsing aims at temporally localizing audio, visual, and audio-visual events in videos and predicting their event categories. Early approaches [1, 2, 31, 32] tried to localize only audio-visual events by aggregating cross-modal information in each local segment of the input video. Due to exhaustive labeling costs on segment-wise audio, visual, and audio-visual events, recent work [3, 4] explored the weakly-supervised audio-visual video parsing with only video-level activity categories for training. For example, HAN [3] introduced the multi-modal multiple instance learning mechanism with a hybrid attention network to aggregate segment-wise representations and video-level features. To alleviate the harm brought by the audio-visual asynchrony, MA [4] proposed the audio-visual contrastive learning from different frames and refined modality-aware labels by exchanging audio and visual tracks between unrelated videos. More recently,

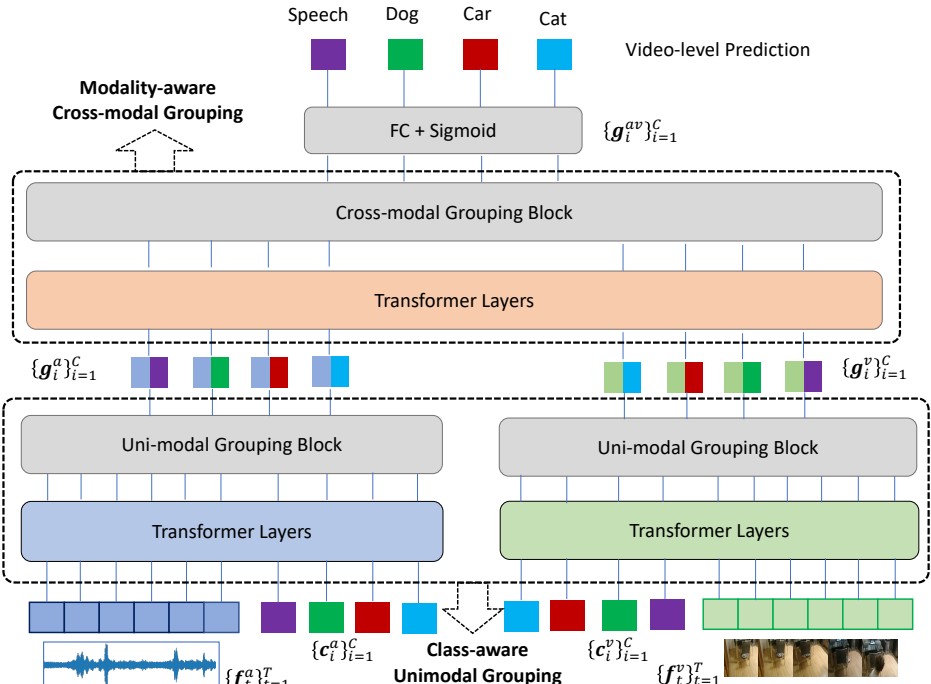

Figure 1: Illustration of our Multi-modal Grouping Network (MGN). The Class-aware Uni-modal Grouping module takes as input raw features $\{\mathbf{f}_t^a\}_{t=1}^T, \{\mathbf{f}_t^v\}_{t=1}^T$ and learnable class tokens $\{\mathbf{c}_i^a\}_{i=1}^C, \{\mathbf{c}_i^v\}_{i=1}^C$ of audio and visual events for $C$ categories to generate class-aware embeddings $\{\mathbf{g}_i^a\}_{i=1}^C, \{\mathbf{g}_i^v\}_{i=1}^C$. The aggregated class-aware representations are fed into the Modality-aware Cross-modal Grouping module to merge all the audio features with video features into new cross-modal modality-aware features $\{\mathbf{g}_i^{av}\}_{i=1}^C$. Finally, an FC layer and sigmoid function are used to predict the video-level target of audio-visual events.

Lin *et al.* [5] leveraged the common semantics shared by videos with replaced audios or frames in a mini batch to detect relevant events.

Different from these baselines based on HAN, we develop a fully novel network architecture to alleviate implicit audio-visual matching and modality category uncertainty in the hybrid attention network. We are the first to exploit unimodal grouping for learning audio-visual representations with class-aware semantics. Furthermore, we introduce a modality-aware cross-modal grouping module to match the video-level label, although the given target does not indicate modalities.

## 3 Method

Given a video with both audio and visual tracks, our goal is to parse the video into temporal segments associated with audible, visible, and audio-visible event categories. We propose a novel baseline: Multi-modal Grouping Network (MGN) to address the task, which mainly consists of two modules: class-aware unimodal grouping (Sec. 3.2) and modality-aware cross-modal grouping (Sec. 3.3).

### 3.1 Preliminaries

In this section, we first describe the problem setup and notations and revisit Multi-modal Multiple Instance Learning in HAN [3] for the audio-visual video parsing (AVVP) task.

**Problem Setup and Notations.** Given a video with $T$ non-overlapping audio and visual segments, our goal is to temporally localize and recognize audio, visual, and audio-visual events that existed in the video. For the multi-label events with $C$ event categories at time $t$, we have audio, visual, and audio-visual event labels for evaluation, which are denoted as: $\mathbf{y}_t^a, \mathbf{y}_t^v, \mathbf{y}_t^{av} \in \mathbb{R}^{1 \times C}$. During training, we do not have the segment- and modality-level annotations. Therefore, we can only use the video-level label $\mathbf{y}^{av} \in \mathbb{R}^{1 \times C}$ to perform weakly-supervised learning.

**Revisit Multimodal Multiple Instance Learning.** To address the weakly-supervised audio-visual video parsing problem, HAN [3] introduced a Multimodal Multiple Instance Learning (MMIL) framework based on transformers [33]. Given a set of audio-visual features $\mathbf{F}^a = \{\mathbf{f}_t^a\}_{t=1}^T, \mathbf{F}^v = \{\mathbf{f}_t^v\}_{t=1}^T$ in $T$ segments, HAN applied self-attention and cross-attention layers to aggregate the unimodal and cross-modal information at each timestamp:

$$\hat{\mathbf{f}}_t^a = \mathbf{f}_t^a + \phi_{sa}(\mathbf{f}_t^a, \mathbf{F}^a, \mathbf{F}^a) + \phi_{ca}(\mathbf{f}_t^a, \mathbf{F}^v, \mathbf{F}^v), \tag{1}$$

$$\hat{\mathbf{f}}_t^v = \mathbf{f}_t^v + \phi_{sa}(\mathbf{f}_t^v, \mathbf{F}^v, \mathbf{F}^v) + \phi_{ca}(\mathbf{f}_t^v, \mathbf{F}^a, \mathbf{F}^a), \tag{2}$$

where $\phi_{sa}(\cdot), \phi_{ca}(\cdot)$ denote the self-attention and cross-attention functions:

$$\phi_{sa}(\mathbf{f}_t^a, \mathbf{F}^a, \mathbf{F}^a) = \text{Softmax}(\frac{\mathbf{f}_t^a \mathbf{F}^{a\top}}{\sqrt{d}})\mathbf{F}^a, \tag{3}$$

$$\phi_{ca}(\mathbf{f}_t^a, \mathbf{F}^v, \mathbf{F}^v) = \text{Softmax}(\frac{\mathbf{f}_t^a \mathbf{F}^{v\top}}{\sqrt{d}})\mathbf{F}^v, \tag{4}$$

and $\mathbf{f}_t^a, \mathbf{f}_t^v \in \mathbb{R}^{1\times d}, \mathbf{F}^a, \mathbf{F}^v \in \mathbb{R}^{T\times d}, d$ is the dimension of audio-visual features. Then, the probability of segment-wise categories for audio and visual events ($\mathbf{p}_t^a, \mathbf{p}_t^v \in \mathbb{R}^{1\times C}$) is predicted by a shared fully-connected (FC) layer and sigmoid function. With the attentive MMIL pooling, the video-level prediction is formulated as:

$$\mathbf{p}^a = \sum_{t=1}^T \mathbf{w}_t^a \mathbf{p}_t^a, \quad \mathbf{p}^v = \sum_{t=1}^T \mathbf{w}_t^v \mathbf{p}_t^v, \quad \mathbf{p}^{av} = \sum_{t=1}^T \sum_{m=1}^M \mathbf{W}_t[m] \odot \mathbf{P}_t[m] \tag{5}$$

where $\mathbf{W}_t = \{\mathbf{w}_t^a, \mathbf{w}_t^v\}$ is the temporal attention weights computed by a FC layer and normalized by a softmax function. $\mathbf{P}_t = \{\mathbf{p}_t^a, \mathbf{p}_t^v\}$ is the probability set of audio-visual predictions. $M = 2$ denotes audio and visual modalities. Finally, the model is trained to optimize a weakly-supervised loss of $\mathbf{p}^{av}$ and a guided loss of $\mathbf{p}^a, \mathbf{p}^v$ with label smoothing:

$$\mathcal{L}_{base} = \text{O}(\mathbf{y}^{av}, \mathbf{p}^{av}) + \text{O}(\overline{\mathbf{y}}^a, \mathbf{p}^a) + \text{O}(\overline{\mathbf{y}}^v, \mathbf{p}^v) \tag{6}$$

where $\text{O}(\cdot)$ is a binary cross-entropy function-based loss term, which summarizes binary cross-entropy for all categories and $\text{O}(\mathbf{y}, \mathbf{p}) = \sum_i BCE(y_i, p_i)$. $\overline{\mathbf{y}}^a, \overline{\mathbf{y}}^v$ are video-level audio and visual labels generated by smoothing $\mathbf{y}^{av}$ to decrease the confidence of positive labels.

However, such a training mechanism will pose two main challenges. First, these methods without explicit grouping suffer from false predictions due to the modality and temporal uncertainties. Second, there is no constraint for modality category prediction to match the video-level target at the end. To address these challenges, inspired by [34], we propose a novel Multi-modal Grouping Network (MGN) with class-aware unimodal grouping and modality-aware cross-modal grouping modules, as shown in Figure 1.

## 3.2 Class-aware Unimodal Grouping

In order to explicitly group class-aware matching semantics for audio-visual representations, we introduce a novel class-aware unimodal grouping module by incorporating learnable modality-specific class tokens $\{\mathbf{c}_i^a\}_{i=1}^C, \{\mathbf{c}_i^v\}_{i=1}^C$ to help to group raw input unimodal features $\{\mathbf{f}_t^a\}_{t=1}^T, \{\mathbf{f}_t^v\}_{t=1}^T$.

We first use self-attention transformers: $\phi_{sa}^a(\cdot)$ and $\phi_{sa}^v(\cdot)$ to temporally aggregate unimodal features from audio and visual inputs and align the features with the categorical token embeddings:

$$\{\hat{\mathbf{f}}_t^a\}_{t=1}^T, \{\hat{\mathbf{c}}_i^a\}_{i=1}^C = \{\phi_{sa}^a(\mathbf{x}_j^a, \mathbf{X}^a, \mathbf{X}^a)\}_{j=1}^{T+C}, \mathbf{X}^a = \{\mathbf{x}_j^a\}_{j=1}^{T+C} = [\{\mathbf{f}_t^a\}_{t=1}^T; \{\mathbf{c}_i^a\}_{i=1}^C] \tag{7}$$

$$\{\hat{\mathbf{f}}_t^v\}_{t=1}^T, \{\hat{\mathbf{c}}_i^v\}_{i=1}^C = \{\phi_{sa}^v(\mathbf{x}_j^v, \mathbf{X}^v, \mathbf{X}^v)\}_{j=1}^{T+C}, \mathbf{X}^v = \{\mathbf{x}_j^v\}_{j=1}^{T+C} = [\{\mathbf{f}_t^v\}_{t=1}^T; \{\mathbf{c}_i^v\}_{i=1}^C] \tag{8}$$

where $[\,;\,]$ denotes the concatenation operator. Then, the unimodal grouping blocks $g^a(\cdot), g^v(\cdot)$ take the learned audio and visual event class tokens and aggregated features as inputs to generate class-aware audio and visual embeddings as:

$$\{\mathbf{g}_i^a\}_{i=1}^C = g^a(\{\hat{\mathbf{f}}_t^a\}_{t=1}^T, \{\hat{\mathbf{c}}_i^a\}_{i=1}^C), \quad \{\mathbf{g}_i^v\}_{i=1}^C = g^v(\{\hat{\mathbf{f}}_t^v\}_{t=1}^T, \{\hat{\mathbf{c}}_i^v\}_{i=1}^C) \tag{9}$$

During grouping, we merge all the unimodal features that belong to the same class token into a new unimodal class-aware feature, by computing the similarity matrices $\mathbf{A}^a, \mathbf{A}^v$ between unimodal features and class tokens via a softmax operation formulated as

$$\mathbf{A}^a_{t,i} = \text{Softmax}(W^a_q \hat{\mathbf{f}}^a_t \cdot W^a_k \hat{\mathbf{c}}^a_i), \quad \mathbf{A}^v_{t,i} = \text{Softmax}(W^v_q \hat{\mathbf{f}}^v_t \cdot W^v_k \hat{\mathbf{c}}^v_i) \tag{10}$$

where $W^a_q, W^a_k$ and $W^v_q, W^v_k$ are the weights of the learned linear projections for the features and class tokens of audio and visual modalities, respectively. Based on this similarity, we calculate the class-aware features with the weighted sum of all segment features assigned to that class:

$$
\begin{aligned}
\mathbf{g}^a_i &= g^a(\{\hat{\mathbf{f}}^a_t\}^T_{t=1}, \hat{\mathbf{c}}^a_i) = \hat{\mathbf{c}}^a_i + W^a_o \frac{\sum^T_{t=1} \mathbf{A}^a_{t,i} W^a_v \hat{\mathbf{f}}^a_t}{\sum^T_{t=1} \mathbf{A}^a_{t,i}} \\
\mathbf{g}^v_i &= g^v(\{\hat{\mathbf{f}}^v_t\}^T_{t=1}, \hat{\mathbf{c}}^v_i) = \hat{\mathbf{c}}^v_i + W^v_o \frac{\sum^T_{t=1} \mathbf{A}^v_{t,i} W^v_v \hat{\mathbf{f}}^v_t}{\sum^T_{t=1} \mathbf{A}^a_{t,i}},
\end{aligned}
\tag{11}
$$

where $W^a_o, W^a_v$ and $W^v_o, W^v_v$ denote the learned weights of linear projections for audio and visual modalities, separately. Note that, the audio features $\{\hat{\mathbf{f}}^a_t\}^T_{t=1}$ are merged with discriminative visual features $\{\mathbf{f}^v_t\}^T_{t=1}$ via a similar grouping block with a hard-softmax operation. In order to constrain the independence of each class token $\mathbf{c}^a_i, \mathbf{c}^v_i$, we apply FC layers and softmax operation to generate the category probability $\mathbf{e}^a_i, \mathbf{e}^v_i$ of class tokens for audio and visual modalities with a class-constrained loss as:

$$\mathcal{L}_{cls} = \sum^C_{i=1} \text{CE}(\mathbf{h}_i, \mathbf{e}^a_i) + \text{CE}(\mathbf{h}_i, \mathbf{e}^v_i) \tag{12}$$

where $\text{CE}(\cdot)$ refers to cross-entropy loss; $\mathbf{h}_i$ is an one-hot encoding vector and only its element for the target class entry $i$ is 1. After the class-aware unimodal grouping, the video-level category predictions $\mathbf{p}^a, \mathbf{p}^v$ of audio and visual events is simply computed by a FC layer and sigmoid operator:

$$\mathbf{p}^a = \text{Sigmoid}(\text{FC}(\{\mathbf{g}^a_i\}^C_{i=1})), \quad \mathbf{p}^v = \text{Sigmoid}(\text{FC}(\{\mathbf{g}^v_i\}^C_{i=1})) \tag{13}$$

With the help of the proposed class-constrained loss, we generate class-aware representations $\{\mathbf{g}^a_i\}^C_{i=1}, \{\mathbf{g}^v_i\}^C_{i=1}$ of audio and visual modalities for audio-visual matching.

### 3.3 Modality-aware Cross-modal Grouping

The second challenge requires us to predict the modality category for matching with the only given video-level target in an explicit way. To achieve this, we propose a modality-aware cross-modal grouping module composed of cross-modal transformers $\phi_{ca}(\cdot)$ and grouping blocks $g^{av}(\cdot)$ to aggregate class-aware representations $\{\mathbf{g}^a_i\}^C_{i=1}, \{\mathbf{g}^v_i\}^C_{i=1}$. Based on the audio-visual similarity in the grouping stage, we combine all the audio features with visual features into new cross-modal modality-aware features $\{\mathbf{g}^{av}_i\}^C_{i=1}$ as:

$$\{\hat{\mathbf{g}}^a_i\}^C_{i=1}, \{\hat{\mathbf{g}}^v_i\}^C_{i=1} = \phi_{ca}([\{\mathbf{g}^a_i\}^C_{i=1}; \{\mathbf{g}^v_i\}^C_{i=1}]) \tag{14}$$

$$\{\mathbf{g}^{av}_i\}^C_{i=1} = g^{av}(\{\hat{\mathbf{g}}^a_i\}^C_{i=1}, \{\hat{\mathbf{g}}^v_i\}^C_{i=1}) \tag{15}$$

where $g^{av}(\cdot)$ denote the grouping operator similar to $g^a(\cdot)$ and $g^v(\cdot)$ in Eq. 11. Then, we leverage the joint audio-visual representations $\{\mathbf{g}^{av}_i\}^C_{i=1}$ to predict the video-level target of audio and visual events via a FC layer and sigmoid function as:

$$\mathbf{p}^{av} = \text{Sigmoid}(\text{FC}(\{\mathbf{g}^{av}_i\}^C_{i=1})) \tag{16}$$

The whole model can be optimized in an end-to-end manner in terms of the objective function:

$$\mathcal{L} = \mathcal{L}_{base} + \mathcal{L}_{cls} \tag{17}$$

At inference time, the unimodal class-aware similarity is used to predict the audio, visual, and audio-visual events for each segment $t$:

$$\mathbf{p}^a_t = \mathbf{p}^a \odot \mathbf{A}^a_t, \quad \mathbf{p}^v_t = \mathbf{p}^v \odot \mathbf{A}^v_t, \quad \mathbf{p}^{av}_t = \mathbf{p}^a_t \odot \mathbf{p}^v_t \tag{18}$$

where $\mathbf{p}^a, \mathbf{p}^v \in \mathbb{R}^{1 \times C}$ and $\mathbf{A}^a, \mathbf{A}^v \in \mathbb{R}^{T \times C}$.

Table 1: Quantitative results of weakly-supervised audio-visual video parsing. 'C' and 'R' denote the contrastive learning and label refinement proposed in MA [4], respectively.

| Method | Segment-Level | | | | | Event-Level | | | | |
|---|---|---|---|---|---|---|---|---|---|---|
| | A | V | A-V | Type | Event | A | V | A-V | Type | Event |
| AVE [1] | 47.2 | 37.1 | 35.4 | 39.9 | 41.6 | 40.4 | 34.7 | 31.6 | 35.5. | 36.5 |
| AVSDN [2] | 47.8 | 52.0 | 37.1 | 45.7 | 50.8 | 34.1 | 46.3 | 26.5 | 35.6. | 37.7 |
| HAN [3] | 60.1 | 52.9 | 48.9 | 54.0 | 55.4 | **51.3** | 48.9 | 43.0 | 47.7 | 48.0 |
| MGN (ours) | **60.7** | **55.5** | **50.6** | **55.6** | **57.2** | 51.0 | **52.4** | **44.4** | **49.3** | **49.2** |
| MA [4] (w C) | **61.9** | 53.1 | 49.7 | 54.9 | 56.2 | **52.8** | 49.4 | 43.8 | 48.7 | 49.0 |
| MGN (w C) | 60.6 | **56.7** | **52.5** | **56.6** | **57.4** | 51.4 | **53.2** | **46.4** | **50.3** | **49.4** |
| MA [4] (w R) | 59.8 | 57.5 | 52.6 | 56.6 | 56.6 | **52.1** | 54.4 | 45.8 | 50.8 | **49.4** |
| MGN (w R) | **60.0** | **60.6** | **54.0** | **58.2** | **58.2** | 50.3 | **58.4** | **47.9** | **52.2** | 49.1 |
| MA [4] (w C+R) | **60.3** | 60.0 | 55.1 | 58.9 | 57.9 | **53.6** | 56.4 | 49.0 | 53.0 | **50.6** |
| MGN (w C+R) | 60.2 | **61.9** | **55.5** | **59.2** | **58.7** | 50.9 | **59.7** | **49.6** | **53.4** | 49.9 |

# 4 Experiments

## 4.1 Experimental Setup

**Dataset.** The Look, Listen and Parse (LLP) Dataset [3] contains 11,849 YouTube video clips of 10-seconds long from 25 different event categories, such as car, music, cheering, speech, etc. Note that each video includes at least 1s audio or visual events and 7202 video clips are annotated with more than one event categories. We use 10,000 video clips with only video-level event labels for training. Following the official splits [3] of validation and test sets, we develop and test the model on the remaining 1879 videos with the segment-level annotations, *i.e.*, the speech event for audio starts at 1s and ends at 5s.

**Evaluation Metrics.** We follow the prior work [3, 4] and use F-scores to evaluate both segment-level and event-level predictions for audio, visual, and audio-visual events. The segment-level metrics can evaluate snippet-wise event prediction performance. For the event-level metrics, we concatenate positive consecutive segments in the same events and compute F-score based on mIoU=0.5 as the threshold. Type@AV and Event@AV are reported for the overall evaluation of audio-visual video parsing performance. Type@AV is the averaged audio, visual, and audio-visual event evaluation results. Different from Type@AV directly averaging results from different event types, Event@AV considers audio and visual events for each sample.

**Implementation.** Following the data pre-processing in previous work [3], we sample video frames at 8 fps from the 10-second videos with 10 non-overlapping snippets of 1 second. For low-level visual features, we concatenate 2D and 3D visual features extracted by ResNet-152 [35] pre-trained on ImageNet [36] and 3D ResNet [37] pre-trained on Kinetics-400 [38]. We utilize VGGish [39] pre-trained on AudioSet [40] to extract the audio features. The model is trained with Adam [41] optimizer with $\beta_1$=0.9, $\beta_2$=0.999 and with an initial learning rate of 3e-4. We train the model with a batch size of 16 for 40 epochs.

## 4.2 Comparison to Prior Work

In this work, we propose a novel and effective training framework for weakly-supervised audio-visual video parsing. To demonstrate the effectiveness of our approach, we first compare it to previous network baselines [2]: 1) AVE [1]: an audio-guided co-attention network with additional audio-visual parsing branches; 2) AVSDN [2]: a sequence-to-sequence-based model with additional audio-visual parsing branches to merge global audio-visual features into local ones; 3) HAN [3]: a hybrid attention network with multi-modal multiple instance learning pooling. Furthermore, we compare MA [4], a method based on HAN with the audio-visual contrastive learning (C) and the label refinement

---

[2]Note that we will not include the recent work [5] into comparison since there is no source code for it. Meanwhile, they are based on HAN [3], but our MGN is a fully novel backbone.

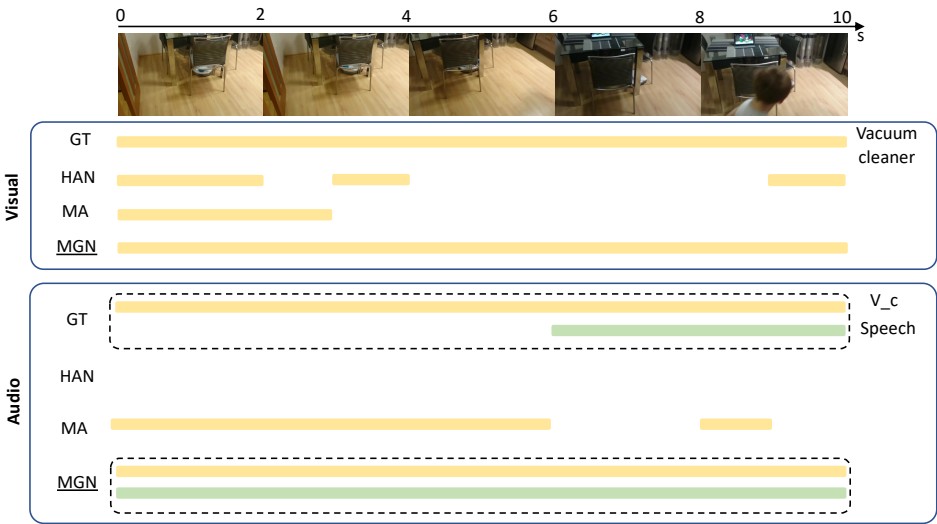

Figure 2: Qualitative comparisons with baselines. "V_c" denotes the "Vacuum_cleaner" class. The proposed MGN achieves much better performance in parsing audio and visual events. Note that the audio-visual event prediction is an intersection of audio and visual event predictions.

(R). The MA series baselines include MA (w C), MA (w R), and MA (w C+R). The quantitative comparisons on the LLP dataset are reported in Table 1.

As can be seen, the proposed MGN achieves the overall best results against previous network baselines in terms most of metrics. For the overall evaluation of segment-level predictions, we achieve significant performance gains of 1.6 Type@AV and 1.8 Event@AV. When evaluated on segment-level predictions of each sample, our MGN also improves the baseline by large margins, 2.6 Visual and 1.7 Audio-Visual. Meanwhile, our MGN outperforms baselines by 3.5 Visual, 1.4 Audio-Visual, and 1.6 Tyep@AV for event-level predictions. These results demonstrate the effectiveness of our approach in weakly-supervised audio-visual video parsing against prior network architectures.

Furthermore, significant gains can be observed in the setting of using the audio-visual contrastive learning and label refinement. Adding the contrastive learning to our MGN achieves the segment-level performance gain of 3.6 Visual and 2.8 Audio-Visual, and the event-level gain of 3.8 Visual and 2.6 Audio-Visual. With the label refinement in MA, we significantly improve MA (w R) by 3.1 segment-level and 4.0 event-level for visual predictions. Our framework with both contrastive learning and label refinement achieves the best segment-level performance in terms of Visual, Audio-Visual, Type@AV, and Event@AV. These improvements imply the strong generalizability of the proposed MGN to the audio-visual contrastive learning and the label refinement.

In order to qualitatively evaluate the predictions of audio and visual events, we compare the proposed MGN with HAN [3], MA [4] in Figure 2. We can observe three main things from previous approaches: First, existing methods can miss the snippet-wise predictions for some hard segments in the video, e.g., MA does not predict the vacuum cleaner for visual events after 3s. Second, prior work can miss one class prediction if there are two categories happening in the same modality. Third, the previous method can miss one modality prediction for the class existing in both modalities. When it comes to the proposed MGN, we superiorly achieve high F1 scores of segment-level and event-level prediction, benefiting from the well-designed class-aware unimodal grouping and modality-aware cross-modal grouping modules.

### 4.3 Experimental Analysis

In this section, we conducted ablation studies to validate the benefit of Class-aware Unimodal Grouping (CUG) and Modality-aware Cross-modal (MCG) Grouping strategies. We also performed extensive experiments to investigate the false positive issue in HAN and the learned class tokens.

Table 2: Ablation studies on Class-aware Unimodal Grouping (CUG) and Modality-aware Cross-modal Grouping (MCG) blocks. Segment-level audio-visual video parsing results are reported.

| CUG | MCG | Audio | Visual | Audio-Visual | Type@AV | Event@AV |
|-----|-----|-------|--------|--------------|---------|----------|
| ✗ | ✗ | 60.1 | 52.9 | 48.9 | 54.0 | 55.4 |
| ✓ | ✗ | 58.9 | 55.3 | 49.8 | 54.7 | 55.9 |
| ✓ | ✓ | **60.7** | **55.5** | **50.6** | **55.6** | **57.2** |

**Class-aware Unimodal Grouping & Modality-aware Cross-modal Grouping.** In order to demonstrate the effectiveness of the proposed class-aware unimodal grouping (CUG) and modality-aware cross-modal grouping (MCG), we ablated the necessity and strategy of grouping blocks. The results of segment-level predictions are reported in Table 2. We can observe that adding CUG to the vanilla baseline achieves significant gains of 2.4 Visual, indicating the effectiveness of grouping class-aware semantics in predicting snippet-wise categories for visual events. Incorporating MCG with CUG highly increases Audio-Visual, Tyep@AV, Event@AV by 1.7, 1.6 and 1.8. These results show the importance of modality-aware grouping on predictions of audio-visual events. Besides effectiveness, our model is also more efficient. When the depth of CUG and MCG is 3 and 6, the proposed MGN with only 47.2% parameters of the vanilla baseline performs the best on Type@AV and Event@AV, especially on Audio. These results further show the advantage of our MGN in real applications with lightweight parameters against the prior work [3, 4]. The detailed results are in Appendix.

**False Predictions.** In order to demonstrate the effectiveness of the proposed MGN in mitigating false predictions against baselines, we calculate the total amount of false positives for all 25 classes in the test set. The comparison results of event-level audio, visual and audio-visual metrics are shown in Figure 3. We can observe that our MGN with the class-aware unimodal grouping modules decreases the false positives of audio and visual events by large margins, 381 and 494. Furthermore, the number of false positives of audio-visual events drops down by 678, which verifies the importance of modality-aware cross-modal grouping in mitigating the modality uncertainty. Overall, our MGN with explicit

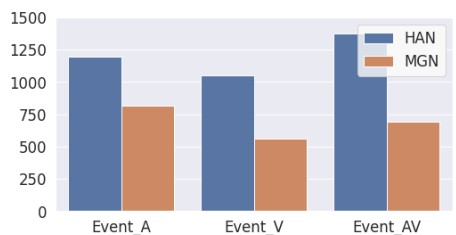

Figure 3: Comparison results of the total amount of false positives for all 25 classes between HAN [3] and the proposed MGN in terms of event-level audio, visual and audio-visual metrics, *i.e.*, Event_A, Event_V, and Event_AV.

grouping mechanisms significantly eliminates false predictions caused by the modality and temporal uncertainties existing in the baseline.

**Learned Class-aware Features.** The learned class tokens are essential to grouping class-aware semantics from audio and visual features. To better evaluate the quality of those learned class-level features, we visualize the learned audio and visual representations of 25 categories by t-SNE [42], as shown in Figure 4. It is noted that each spot denotes the feature of one audio or visual event, while each color represents each class, such as "Speech" in brown and "Dog" in green. As can be seen in the last column, features extracted by the proposed MGN are intra-class compact and inter-class separable. However, there still exists mixtures of multiple categories for audio and visual events among the representations of HAN and MA. For the sub-figure on the bottom right, we can observe a large cluster of brown spots for the "Speech" class of audio events in the test set, while brown spots in prior work are distributed more sparsely. These meaningful visualizations further demonstrate that our MGN successfully learns compact and discriminative features for each modality.

## 4.4 Limitation

Although the proposed MGN achieves superior results on visual events and audio-visual events, the gains of audio events are not significant compared to the visual modality. We notice that there are 1628 visual instances and 2663 audio instances in the test set, which implies the audio modality is much harder than the visual one in this setting. One possible solution is to introduce a small number of segment-wise audio-visual parsing annotations as supervision for semi-supervised training. In the

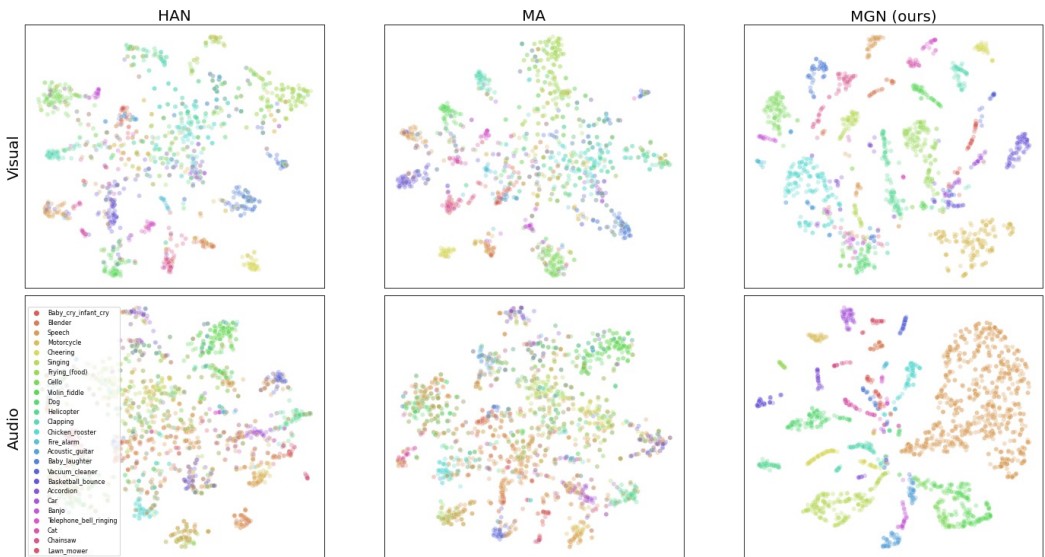

Figure 4: Qualitative visualizations of audio (Top rows) and visual (Bottom rows) features learned by HAN, MA and the proposed MGN. Note that each spot denotes the feature of one audio or visual event, while each color represents each class, such as "Speech" in brown and "Dog" in green.

meanwhile, our MGN performs worse with the increase of the depth of transformer layers in grouping modules. This is caused by such a weakly-supervised setting with only video-level annotations that do not indicate either segments or modalities. However, the model is expected to parse a video into events with different categories and modalities. Therefore, the potential future work is to add more grouping stages with learned class-tokens as supervision for each one.

## 5   Conclusion

In this work, we present MGN, a fully novel Multi-modal Grouping Network to explicitly group class-aware matching semantics for weakly-supervised audio-visual video parsing. We introduce the class-aware unimodal grouping module to generate class-aware unimodal representations with learnable tokens by using unimodal grouping blocks for each modality. Furthermore, we leverage the modality-aware cross-modal grouping to match the video-level target with the cross-modal grouping blocks. Experimental results demonstrate the effectiveness and superiority of our MGN against previous baselines. We also show the generalizability of our simple framework to the audio-visual contrastive learning and label refinement.

**Broader Impact.** The proposed method detects video events in audio and visual modalities based on the learned statistics of the training dataset. It could capture internal biases in the data, which may have negative societal impacts. For example, the model might not be able to discover rare but important events, such as fire warnings. Thus, before deploying our audio-visual video parsing model into real-world applications, we need to carefully address these issues.

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
