# Multi-modal Grouping Network for Weakly-Supervised Audio-Visual Video Parsing
## *(Supplementary Material)*

**Shentong Mo**
Carnegie Mellon University

**Yapeng Tian**[*]
University of Texas at Dallas

In the supplementary material, we provide the differences between our MGN and the recent work, GroupViT [1], more experiments on model size and grouping strategies. We finally validate the effectiveness of audio/visual tokens in learning class-aware embeddings. The code of our implementation is also attached as the supplementary material for review.

## 1   Differences between GroupViT [1] and the proposed MGN

Compared to GroupViT [1], the recent grouping work on image segmentation, there are three main distinctive characteristics of our MGN for solving the weakly-supervised auido-visual video parsing problem, which are listed as follows:

1) **Constraint on Class Tokens.** Most important, we have learned 25 category tokens with each class constraint, *i.e.*, each class token does not have semantic overlapping information among each other and we want each token to learn the embeddings for each category. To achieve this, we apply the cross-entropy loss to class tokens for audio-visual modalities as $\mathcal{L}_{cls} = \text{CE}(\{\mathbf{p}_i^a\}_{i=1}^C, \mathbf{1}) + \text{CE}(\{\mathbf{p}_i^v\}_{i=1}^C, \mathbf{1})$. However, the number of learned group tokens in GroupViT is a hyper-parameter and there is no constraint on it.

2) **Modality-aware Cross-modal Grouping.** We have the modality-aware cross-modal grouping to eliminate the modality uncertainty caused by the weakly-supervised setting in the task, while GroupViT does not incorporate text embeddings into the grouping stage. The text embeddings is used in a contrastive loss to match with the global visual representations. In this case, their model was trained on large-scale data with a large batch size for self-supervised training.

3) **Class-aware Unimodal Grouping.** We introduce the class-aware unimodal grouping for mitigating the temporal uncertainty in each modality, but GroupViT only leverages the unimodal grouping on visual patch tokens without class tokens involved. Therefore, the unimodal grouping in GroupViT can not be used in this task directly. Furthermore, they apply multiple grouping stages during training and this is also a hyper-parameter. In our case, one grouping stage is enough for us to learn discriminative representations in the embedding space with the help of meaningful class tokens.

## 2   Depth of Transformer Layers and Model Size

The depth of transformer layers affects the extracted audio-visual representations for grouping and the training parameters as well. To explore such effect more comprehensively, we varied the depth of CUG and MCG modules from $\{1, 2, 3, 4, 6, 8\}$. Table 1 reports the comparison results of the segment-level performance and the number of parameters. When the depth of CUG and MCG is 1 and 2, the proposed MGN with the least parameters 11 MB increases the baseline with parameters 36 MB by 1.7 Visual and 1.1 Event@AV. With the increase of the depth of CUG, we achieve consistently improving performance of Visual and Audio-Visual for segment-level prediction since we extract better visual representations from transformers for class-aware unimodal grouping. When the depth

---

[*]Corresponding author.

36th Conference on Neural Information Processing Systems (NeurIPS 2022).

Table 1: Exploration study on model sizes and the depth of transformer layers in Class-aware Unimodal Grouping (CUG) and Modality-aware Cross-modal Grouping (MCG) modules. Segment-level audio-visual video parsing results are reported.

| Method | CUG depth | MCG depth | Model Size | Audio | Visual | Audio-Visual | Type@AV | Event@AV |
|--------|-----------|-----------|------------|-------|--------|--------------|---------|----------|
| HAN | N/A | N/A | 36 MB | 60.1 | 52.9 | 48.9 | 54.0 | 55.4 |
| MGN | 1 | 2 | **11 MB** | 59.7 | 54.6 | 49.2 | 54.5 | 56.5 |
| MGN | 2 | 4 | 14 MB | 58.9 | 55.2 | 50.2 | 54.8 | 55.7 |
| MGN | 3 | 6 | 17 MB | **60.7** | 55.5 | 50.6 | **55.6** | **57.2** |
| MGN | 4 | 8 | 20 MB | 60.1 | 54.2 | 50.0 | 54.8 | 56.2 |
| MGN | 3 | 3 | 15 MB | 58.7 | 55.3 | 49.8 | 54.6 | 55.8 |
| MGN | 6 | 3 | 20 MB | 59.0 | **55.6** | **50.7** | 55.1 | 55.9 |

Table 2: Ablation studies on Class-aware Unimodal Grouping (CUG) and Modality-aware Cross-modal Grouping (MCG) blocks. Segment-level audio-visual video parsing results are reported.

| CUG | MCG | Audio | Visual | Audio-Visual | Type@AV | Event@AV |
|-----|-----|-------|--------|--------------|---------|----------|
| Hard-Softmax | Softmax | 59.9 | 54.9 | **50.4** | 55.1 | 56.2 |
| Hard-Softmax | Hard-Softmax | 57.5 | 54.9 | 50.3 | 54.2 | 54.8 |
| Softmax | Hard-Softmax | 59.7 | 54.5 | 49.6 | 54.6 | 56.2 |
| Softmax | Softmax | **60.7** | **55.5** | **50.6** | **55.6** | **57.2** |

of CUG and MCG is 3 and 6, the proposed MGN with only 47.2% parameters of the vanilla baseline performs the best on Type@AV and Event@AV, especially on Audio. These results further shows the advantage of our MGN in real applications with lightweight parameters against the prior work [2, 3].

## 3  Grouping Strategy

In addition, we ablated the strategy of CUG and MCG using Hard-Softmax and Softmax. To make Hard-Softmax differentiable during training, we adopted the Gumbel-Softmax [4, 5] as the alternative. Table 2 reports the ablation results. With Softmax as both unimodal and cross-modal grouping strategies, the proposed MGN achieves the best performance. Replacing the Softmax with the hard version in either unimodal or cross-modal grouping deteriorate the segment-level predictions, which also complies to the multi-label property of each segment for both modalities.

## 4  More Analysis on Recall

In order to figure out the reduction of false positives, we compare the recall for all 25 classes in the test set. The comparison results of event-level audio, visual and audio-visual metrics are shown in Figure 1. We can observe that the reduction of false positives by the proposed MGN over baselines indeed comes at the cost of a decrease in recall for audio, visual or audio-visual events.

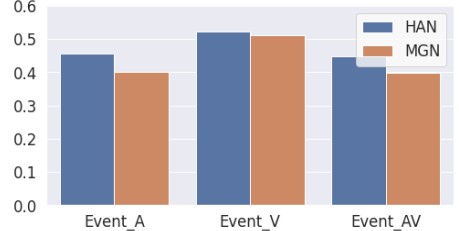

Figure 1: Comparison results of recall for all 25 classes between HAN [2] and the proposed MGN in terms of event-level audio, visual and audio-visual metrics, *i.e.*, Event_A, Event_V, and Event_AV.

## 5  Quantitative Validation on Audio/Visual Class Tokens

In order to quantitatively validate the rationality of learned audio/visual token embeddings, we compute the Precision, Recall, and F1 scores of those tokens during training. The quantitative results

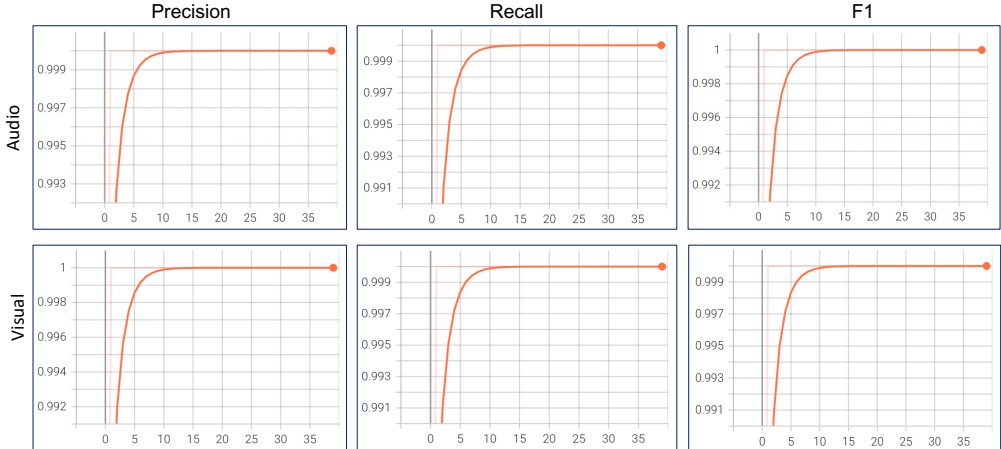

Figure 2: Quantitative results (Precision, Recall, and F1 score) of learned audio and visual class tokens.

are shown in Figure 2. We can observe that all metrics arrive to closer to 1 at epoch 10, which means that the learned class tokens are semantically class-aware. Furthermore, these results demonstrate the effectiveness of class-aware tokens in the class-aware unimodal grouping for alleviating the temporal uncertainty in each modality.