# OpenReview forum: "Multi-modal Grouping Network for Weakly-Supervised Audio-Visual Video Parsing"
_NeurIPS.cc/2022/Conference — NeurIPS 2022 Accept_

### Official Review · Reviewer_f8HF · 2022-07-13

**Rating:** 6
**Confidence:** 4
**Soundness:** 3 good
**Presentation:** 2 fair
**Contribution:** 3 good

**Summary:**

This paper addresses the problem of predicting event labels over time in audio visual data with weak labels.  A method is proposed that uses attention between audio visual features and learned class embeddings, and extracts class-specific embeddings for use in event detection.  The model is evaluated on the Look, LIsten, and Parse dataset, using 11K video clips with video labels for training, and around 2K video clips with both audio and video labels for evaluation.

**Questions:**

See Weaknesses above for some questions.

**Ethics Review Area:**

["I don’t know"]

**Limitations:**

The authors acknowledge that performance on audio events is not improved.

**Strengths And Weaknesses:**


Strengths:
The proposed method is an interesting idea, and seems to help with the task.
Small improvements are achieved on a challenging task evaluated on a sufficiently large test set.

The ablations in Table 2 are informative.

Weaknesses:
The performance on audio events is not improved.

The analysis of false positives in Figure 3 is useful, but it seems like the models are just biased to different operating points, since their F-scores in table 1 are similar.  So the reduction of false positives in Figure 3 must come at the cost of a decrease in recall.   It would be better to show both numbers, or calibrate to the same operating point.


Clarity:

The paper is a bit difficult to understand due to notational issues

> These methods without explicit grouping suffer from false predictions due to the modality and temporal uncertainties.

It is not clear to me why that should be at this point in the paper.  Is this an empirical finding or a hypothesis?

Re: equation (6), cross entropy is usually from labels to estimates, so I would expect $CE(p,\\hat{p})$, but here it seems to be used the reverse way.  Also this looks like a vectorized version with binary probabilities in the elements.  It might be good to define it since technically cross entropy is defined over a distribution, rather than a vector of distributions.  That is, I think what is meant is something like $CE(\\mathbf{y}, \mathbf{p}) = \\sum_i CE(y_i, p_i)$.  If not please clarify.

In the cross-modal attention $\\phi_{ca}(q,K,V)$, my understanding is that the key and value should be of the same modality, and the query of the other modality.
So if I understand correctly, in equation (1)  $\\phi_{ca}(f^v_t, F^v, F^a)$  is probably supposed to be $\\phi_{ca}(f^a_t, F^v, F^v)$ , and in equation (2) $\\phi_{ca}(f^v_t, F^a, F^v)$  should be $\\phi_{ca}(f^v_t, F^a, F^a)$.

From equation (7) on, the notation for self-attention changes without warning from a three-argument function to a one argument function $\\phi_{sa}(X)$, which I assume in the notation of (3) would be $\\{\\phi_{sa}(x_i, X, X)\\}_i$, but it would be kinder to the reader to just define it.

In equation (10), it is not clear why learned weights are needed to transform both the class tokens and the modality specific features.  Is it not equivalent to just transform the features?  That is $Ax \\cdot By = x^T A^T B y = B^T A x \\cdot  y = W x \\cdot y $, where $W = B^T A$.

In equation (12), the notation is not clear.   The $\\mathbf{1}$ is not clear to me.  I was expecting to see a label for the presence of the class.  I'm probably misunderstanding something, but in any case, I think this needs to be better explained. Perhaps writing out the CE formula element by element would clarify.   Also, maybe (13) should come before (12) for clarity.


Errata:

> In order to explicitly group~~ing~~ class-aware...

---

> ### Author Response · Authors · 2022-08-02
> **Response to R3**
>
> **W1**
> *The performance on audio events is not improved.*
>
> As we discussed in Sec 4.4, although the proposed MGN achieves superior results on visual events and audio-visual events, the gains of audio event parsing are not significant compared to the visual modality. We observed that audio content is much nosier in videos. Different visual objects are naturally separated in video frames. However, multiple audio sources can be mixed together at the same time, and not all of the sources are visually appeared, which makes it challenging to disentangle noise-free features for grouping. Thus, to improve the new multi-modal grouping-based architecture, one promising direction is to further enforce separating individual audio sources in the grouping mechanism.
>
>
> **W2**
> *The analysis of false positives in Figure 3 is useful, but it seems like the models are just biased to different operating points, since their F-scores in table 1 are similar. So the reduction of false positives in Figure 3 must come at the cost of a decrease in recall. It would be better to show both numbers, or calibrate to the same operating point.*
>
> Thanks for the suggestion! We have added the comparison of recall to Sec.D in the appendix.
> The reduction of false positives by the proposed MGN over baselines indeed comes at the cost of a decrease in recall for audio, visual or audio-visual events.
>
> **Q1**
> *The paper is a bit difficult to understand due to notational issues.*
>
> We have proofed each line and fixed notational issues below in the revision.
>
>
> **Q2**
> *"These methods without explicit grouping suffer from false predictions due to the modality and temporal uncertainties."
> It is not clear to me why that should be at this point in the paper. Is this an empirical finding or a hypothesis?*
>
> This is an empirical finding obtained after looking at results of previous models (e.g., HAN [3]) on the LLP dataset.
>
>
> **Q3**
> *Re: Equation (6), cross entropy is usually from labels to estimates, so I would expect $CE(p,\hat{p})$, but here it seems to be used the reverse way. Also this looks like a vectorized version with binary probabilities in the elements. It might be good to define it since technically cross entropy is defined over a distribution, rather than a vector of distributions. That is, I think what is meant is something like $CE(\mathbf{y},\mathbf{p}) = \sum_{i}CE(y_i,p_i)$. If not please clarify.*
>
> Since it is a multi-label multi-class problem, the CE($\cdot$) in Equation (6) is a binary cross-entropy function-based loss term, which summarizes binary cross-entropy for all categories.
> Thus, $CE(\mathbf{y},\mathbf{p}) = \sum_{i}BCE(y_i,p_i)$.
> To avoid confusion, we replaced the CE($\cdot$) with $\mbox{O}(\cdot)$ in Equation (6).
> We have clarified it in the revision (Line 123-125).
>
>
> **Q4**
> *In the cross-modal attention $\phi_{ca}(q, K, V)$, my understanding is that the key and value should be of the same modality, and the query of the other modality.
> So if I understand correctly, in equation (1) $\phi_{ca}(f_t^{v}, F_v, F_a)$ is probably supposed to be $\phi_{ca}(f_t^{a}, F_v, F_v)$, and in equation (2) $\phi_{ca}(f_t^{a}, F_a, F_v)$ should be $\phi_{ca}(f_t^{v}, F_a, F_a)$.*
>
> Thanks for pointing this out.
> Your understanding is correct, as we defined in Equation (4) for $\phi_{ca}(f_t^{a}, F_v, F_v)$.
> This is a typo, and we have fixed it in the revision (Line 113).
>
>
> **Q5**
> *From equation (7) on, the notation for self-attention changes without warning from a three-argument function to a one argument function , which I assume in the notation of (3) would be $\{\phi_{sa}(x_i, X, X)\}_{i}$, but it would be kinder to the reader to just define it.*
>
> Thanks for your kind suggestion!
> We have modified it in the revision (Line 138-139).

---

> > ### Author Response · Authors · 2022-08-02
> > **Response to R3 (cont.)**
> >
> > **Q6**
> > *In Equation (10), it is not clear why learned weights are needed to transform both the class tokens and the modality specific features. Is it not equivalent to just transform the features? That is $Ax\cdot By = x^TA^TBy = B^TAx\cdot y = Wx\cdot y$, where $W=B^TA$.*
> >
> > Yes, they are theoretically equivalent.
> > Here, we just adopt a common practice that uses individual learnable linear projections for transforming query and key in the multi-head attention operation.
> > In our CUG block, query and key refer to the features and class tokens of each individual modality.
> >
> >
> > **Q7**
> > *In Equation (12), the notation is not clear. The $\mathbf{1}$ is not clear to me. I was expecting to see a label for the presence of the class. I'm probably misunderstanding something, but in any case, I think this needs to be better explained. Perhaps writing out the CE formula element by element would clarify. Also, maybe (13) should come before (12) for clarity.*
> >
> > In Equation (12), $\mathbf{1}$ should be one-hot encoding and only its element for the target entry $i$ is 1.
> > We follow your suggestion and re-write the formula element by element as
> > $\mathcal{L}_{cls} = \sum_i \mbox{CE}(\mathbf{h}_i, \mathbf{e}^a_i)+\mbox{CE}(\mathbf{h}_i, \mathbf{e}^v_i)$,
> > where $\mbox{CE}(\cdot)$ refers to cross-entropy loss; $\mathbf{h}_i$ is an one-hot encoding vector and only its element for the target class entry $i$ is 1;
> > $\mathbf{e}^a_i, \mathbf{e}^v_i$ are category predictions of learned tokens for each individual modality.
> >
> > In Equation (13),
> > $\mathbf{p}^a, \mathbf{p}^v$ are video-level event category predictions for audio and visual inputs of the video. Thus, there is no a direct relation between Equation (12) and Equation (13).
> > Previously, we used $\mathbf{p}_i^{a}$ and $\mathbf{p}_i^{v}$ to denote the token class probabilities, which causes the concern.
> > Now we have used $\mathbf{e}$ to replace $\mathbf{p}$ in Equation (12).
> >
> > We have clarified these in the revision (Line 152-157).
> >
> >
> > **Q8**
> > *Errata:
> > In order to explicitly group~ing~ class-aware…*
> >
> > We have fixed the typo in the revision (Line 134).

---

### Official Review · Reviewer_kAVj · 2022-07-16

**Rating:** 6
**Confidence:** 3
**Soundness:** 3 good
**Presentation:** 3 good
**Contribution:** 2 fair

**Summary:**

The authors present a new baseline architecture for video parsing. Using learnable categorial embedding tokens they propose class-aware unimodal grouping network in conjunction with a cross-modal grouping network to time-stamp audio, visual and audio-visual events using only video level labels. They show improved results compared to other baselines on LLP dataset.

**Questions:**

- The model uses pre-trained ResNet and VGGish net for model input, both of which are very good feature extractors. What is the importance/requirement of using such feature extractors on the final result. More specifically, LLP dataset is derived from Audioset and VGGish net is trained in a supervised fashion using AudioSet, which makes it an ideal feature extractor. Can the author's comment if the unimodal grouping network would still be effective if not for such feature extractor.
- How does the model perform with event scaling ? (Any reason to limit the study to LLP and not use full AudioSet)?
- Regarding implementation (line 188), How low can the time-resolution go? (ie., instead of 1s snippets, can 0.1s snippets be used?)
- in Table2, why does using CUG only makes “Audio” result worse when compared to not using ?  (row1 vs row2)

**Limitations:**

see *Questions* section.

**Strengths And Weaknesses:**

The manuscript discusses the important problem of video parsing with very coarse labels. It is well written and easy to read through. The limited number of experiments presented are through and insightful. The proposed method if effective compared to other baselines as evident from the qualitative and quantitative results. However, it is heavily inspired from recently proposed GroupViT for image segmentation. Though the authors outline the differences between proposed approach and GroupViT in supplementary material, they are very nuanced and not significant.

---

> ### Author Response · Authors · 2022-08-02
> **Response to R2**
>
> **W1**
> *However, it is heavily inspired from recently proposed GroupViT for image segmentation. Though the authors outline the differences between proposed approach and GroupViT in supplementary material, they are very nuanced and not significant.*
>
> Yes, our method is inspired by the recent image segmentation method: GroupViT. However, not simply using GroupViT to solve the audio-visual video parsing problem, there are three significant differences between GroupViT and the proposed MGN in the grouping mechanism design as mentioned in the supplementary material:
>
> 1) **No Constraint on Class Tokens**: the number of learned group tokens in GroupViT is a hyper-parameter and there is no constraint on it. However, we propose to learn categorical tokens with a semantic-aware constraint, i.e., each class token does not have semantic overlapping information among each other.
>
> 2) **No Modality-aware Cross-modal Grouping**:
> we involved the audio modality in the modality-aware cross-modal grouping to eliminate the modality uncertainty caused by the weakly-supervised setting in the task.
> But GroupViT does not incorporate the text modality into the grouping stage and utilizes a contrastive loss to match with the global visual representations.
>
> 3) **No Class-aware Unimodal Grouping**:
> GroupViT only leverages the unimodal grouping on visual patch tokens without class tokens involved, while we introduced the class-aware unimodal grouping with learnable class tokens for mitigating the temporal uncertainty in each individual modality.
>
>
> **Q1**
> *The model uses pre-trained ResNet and VGGish net for model input, both of which are very good feature extractors. What is the importance/requirement of using such feature extractors on the final result. More specifically, LLP dataset is derived from Audioset and VGGish net is trained in a supervised fashion using AudioSet, which makes it an ideal feature extractor. Can the author's comment if the unimodal grouping network would still be effective if not for such feature extractor.*
>
> (1) The proposed MGN is not dependent on a specific audio feature extractor, and there is no assumption on unimodal grouping for using what kinds of audio features.
>
> (2) We want to clarify that the adopted audio feature extractor: VGGish model was trained on YouTube-8M instead of AudioSet (https://github.com/tensorflow/models/blob/master/research/audioset/vggish/README.md). So, the used VGGish model might not be an ideal feature extractor for LLP and the grouping network should still be effective even not using this feature extractor.
> But we do agree with the reviewer that a VGGish trained using the full AudioSet should be more powerful.
>
>
> **Q2**
> *How does the model perform with event scaling? (Any reason to limit the study to LLP and not use full AudioSet)?*
>
> This is a good question!
> Although the AudioSet is a video dataset containing both audio and visual tracks, it was organized and collected only regarding audio content. The dataset only has audio event labels and lacks of visual event annotations (audio and visual events are not always in the same categories in real-world videos).
> For our audio-visual video parsing model, the weak labels should reflect both audio and visual events in videos. So, we need additional visual event annotations for training our network. Besides, there is no segment-level annotations for evaluation.
>
>
> **Q3**
> *Regarding implementation (line 188), How low can the time-resolution go? (ie., instead of 1s snippets, can 0.1s snippets be used?*
>
> Too short audio events are hard for neural networks, even for human beings, to perceive the event category. As suggested in a sound event detection challenge (https://dcase.community/challenge2018/task-large-scale-weakly-labeled-semi-supervised-sound-event-detection), the time-resolution can be lower to 0.25s.
>
>
> **Q4**
> *in Table2, why does using CUG only makes “Audio” result worse when compared to not using ? (row1 vs row2)*
>
>  In the LLP dataset, audio content is noisier compared to the visual modality.
>  So only using CUG may propagate noisy segment-level features.
>  But leveraging the visual information as a condition in MCG boosts the Audio result, which demonstrates the necessity of integrating both CUG and MCG.

---

### Official Review · Reviewer_KNZ9 · 2022-07-18

**Rating:** 4
**Confidence:** 4
**Soundness:** 3 good
**Presentation:** 2 fair
**Contribution:** 2 fair

**Summary:**

This paper tackles weakly-supervised audio-visual video parsing, and proposes a Multi-modal Grouping Network to explicitly group class-aware matching semantics with class-aware unimodal grouping module and modality-aware cross-modal grouping module. Experimental results are shown on the Look, Listen and Parse (LLP) Dataset with the generalizability to the audio-visual contrastive learning and label refinement.

**Questions:**

1. L58: "We propose a new audio-visual video parsing baseline....."     Why is the proposed model called a "baseline"?
2.  For the results in Table 1, the best result for "Event-Level"-"A" is not highlighted. Why?

**Limitations:**

The authors list the model limitations of limited data, and worse performance with deeper network, and point out the possible solutions of semi-supervised training and incorporating more intermediate supervision. The authors also point out the case of rare events in real deployment, which is practical and has corresponding techniques focusing on the case.

**Strengths And Weaknesses:**

- Strengths:
This paper tackles weakly-supervised audio-visual video parsing considering the asynchronous possibility of the two modalities.

- Weaknesses:
1. The introduction of this paper is not strongly motivated, but only with general and brief description. The introduction lists several related papers but the relations of these papers to the proposed model are not well illustrated.

2. L63: "The experiments can demonstrate the superiority of our MGN over state-of-the-art AVVP approaches and its generalizability to contrastive learning and label refinement."     When the authors fist mention "contrastive learning and label refinement" [4] in the introduction, these two concepts are not explained but directly appear with citation to another paper, which makes the paper reading confusing.

3. For the results in "Table 2: Ablation studies on Class-aware Unimodal Grouping (CUG) and Modality-aware Cross-modal Grouping (MCG) blocks, each of CUG block and MCG block is ablated as a whole. What about the effects of the design choices in each block? E.g. the effect of concatenating class labels in Class-aware Unimodal Grouping (CUG) block?

4. Only one dataset is experimented on.

---

> ### Author Response · Authors · 2022-08-02
> **Response to R1**
>
> **W1**
> *The introduction of this paper is not strongly motivated, but only with a general and brief description. The introduction lists several related papers but the relations of these papers to the proposed model are not well illustrated.*
>
> Our key motivation is to learn compact and discriminative audio and visual representations by explicit multi-modal grouping for mitigating the modality and temporal uncertainties in the weakly-supervised audio-visual video parsing problem.
> Existing approaches [3,4,5] usually focus on learning to leverage the unimodal and cross-modal temporal contexts from weak supervisions. During training, segment-wise event labels are unavailable for audio and visual temporal segments in videos. Thus, the multi-modal temporal modeling in these methods might aggregate and implicitly group irrelevant semantic information due to lacking of fine-grained supervision, which causes false positives for predicting categories of events, as shown in Figures 3 and 4. Different from past approaches, we propose a new Multi-modal Grouping Network, namely MGN, to explicitly group semantic-aware multi-modal contexts, which enables learning more compact and discriminative audio and visual representations. We have modified our introduction accordingly (please kindly check the highlighted red texts in the revised paper).
>
>
> **W2**
> *L63: "The experiments can demonstrate the superiority of our MGN over state-of-the-art AVVP approaches and its generalizability to contrastive learning and label refinement." When the authors first mention "contrastive learning and label refinement" [4] in the introduction, these two concepts are not explained but directly appear with citation to another paper, which makes the paper reading confusing.*
>
> Thanks for pointing this out! Contrastive learning and label refinement are proposed in [4], where they adopted a contrastive loss to enforce the temporal alignment between the audio and visual features at the same timestamp and augmented training data with modality-aware event labels generation.
> We have clarified them in the revision (Line 32-35).
>
>
> **W3**
> *For the results in "Table 2: Ablation studies on Class-aware Unimodal Grouping (CUG) and Modality-aware Cross-modal Grouping (MCG) blocks, each of CUG block and MCG block is ablated as a whole. What about the effects of the design choices in each block? E.g. the effect of concatenating class labels in Class-aware Unimodal Grouping (CUG) block?*
>
> 1) Thanks for the suggestion! We explored different design choices inside each block in the appendix, such as depth in Sec.B and grouping strategy in Sec.C.
>
> 2) For the effect of concatenating class labels, this design choice is not feasible.
> This is because we can only access video-level event labels during training, and we do not know which temporal segments contain and which modalities perceive these events.
> If we concatenate video-level event labels in CUG block, those labels leak cross-modal clues in the unimodal grouping process, which deteriorates the discriminativeness and compactness of unimodal representations.
> To learn compact and discriminative audio and visual representations, we extract event-aware unimodal features through unimodal grouping in terms of learnable categorical embedding tokens for each individual modality.
>
>
> **W4**
> *Only one dataset is experimented on.*
>
> To the best of our knowledge, the LLP dataset is the only existing benchmark for the audio-visual video parsing problem.
> But it is large and indeed challenging as it comes from more than 11,000 YouTube video clips of 10-seconds long from 25 different event categories.
>
>
> **Q1**
> *L58: "We propose a new audio-visual video parsing baseline....." Why is the proposed model called a "baseline"?*
>
> Recent audio-visual video parsing methods usually adopt HAN [3] as a baseline and add new modules and losses over it. Our MGN as a novel audio-visual video parsing architecture can flexibly integrate these advanced designs to boost performance. Thus, we call the proposed MGN as a new "baseline".
>
>
> **Q2**
> *For the results in Table 1, the best result for "Event-Level"-"A" is not highlighted. Why?*
>
> In Table 1, we only highlighted the result when our method surpasses the baseline in terms of same-level comparisons, such as 60.8 and 60.0 in "Segment-Level"-"A". To keep the notation consistent, we now highlight all of the best results in the modified Table 1.

---

### Author Response · Authors · 2022-08-02
**Comments to all reviewers**

We thank all the reviewers for carefully reading our paper and providing constructive comments!
We address concerns from the three reviewers as below.

---

### Meta-Review · Area_Chair_ocNi · 2022-08-26

**Recommendation:** Accept
**Confidence:** Certain

**Metareview:**

The authors propose an approach for weakly supervised audio-visual parsing of videos. They propose using learnable categorical embedding to do class-aware unimodal grouping, combined with cross-modal grouping to time-stamp audio, visual and audio-visual events using only video level labels.

Based on the feedback provided by the reviewers, especially since Reviewer KNZ9 increased their score to Borderline Accept after the rebuttal period, we recommend this paper for publication at NeurIPS 2022.

The reviewers had some concerns about the paper. Reviewer KNZ9 mentioned that the relations of the listed papers to the proposed model were not well explained -- they also had some concerns about the experimental results, and the fact that only one dataset was used in the evaluation. Reviewer kAVj had questions about model performance with event scaling, and time resolution lower bounds. Reviewer f8HF commented on the difficulty in following the notation in the paper, and pointed out the results on audio events is not improved.

We thank the authors for addressing the comments of the reviewers in their review during the author feedback period. The authors seem to have addressed some of the concerns/feedback from the reviewers with detailed discussions -- it would be good to include these discussions, as much as possible, in the updated paper or supplemental materials.

**Award:**

No

---

### Decision · Program_Chairs · 2022-09-14

Accept